# Increased formate overflow is a hallmark of oxidative cancer

Johannes Meiser [1], Anne Schuster[2], Matthias Pietzke[1], Johan Vande Voorde[1], Dimitris Athineos [1], Kristell Oizel[1], Guillermo Burgos-Barragan[3], Niek Wit[3], Sandeep Dhayade[1], Jennifer P. Morton[1,4], Emmanuel Dornier [1], David Sumpton[1], Gillian M. Mackay[1], Karen Blyth [1], Ketan J. Patel[3,5], Simone P. Niclou [2,6] & Alexei Vazquez [1,4]

Formate overflow coupled to mitochondrial oxidative metabolism\ has been observed in cancer cell lines, but whether that takes place in the tumor microenvironment is not known. Here we report the observation of serine catabolism to formate in normal murine tissues, with a relative rate correlating with serine levels and the tissue oxidative state. Yet, serine catabolism to formate is increased in the transformed tissue of in vivo models of intestinal adenomas and mammary carcinomas. The increased serine catabolism to formate is associated with increased serum formate levels. Finally, we show that inhibition of formate production by genetic interference reduces cancer cell invasion and this phenotype can be rescued by exogenous formate. We conclude that increased formate overflow is a hallmark of oxidative cancers and that high formate levels promote invasion via a yet unknown mechanism.

[1] Cancer Research UK Beatson Institute, Glasgow G61 1BD, UK. [2] Department of Oncology, NorLux Neuro-Oncology Laboratory, Luxembourg Institute of Health, L-1526 Luxembourg, Luxembourg. [3] MRC Laboratory of Molecular Biology, Cambridge CB2 0QH, UK. [4] Institute for Cancer Sciences, University of Glasgow, G61 1BD Glasgow, UK. [5] Department of Medicine, Addenbrooke's Hospital, University of Cambridge, Cambridge CB2 2QQ, UK. [6] Department of Biomedicine, Kristian Gerhard Jebsen Brain Tumour Research Center, University of Bergen, Bergen N-5009, Norway. These authors contributed equally: Anne Schuster, Matthias Pietzke, Johan Vande Voorde. Correspondence and requests for materials should be addressed to A.V. (email: a.vazquez@beatson.gla.ac.uk)

n his 1956 landmark paper Otto Warburg hypothesized that cancer is caused by mitochondrial defects that result in increased rates of glycolysis with lactate overflow[1]. Today increased glycolysis is an established hallmark of cancer metabolism and forms the scientific basis for Positron Emission Tomography (PET) scans. In contrast, the Warburg hypothesis that cancers harbor defective mitochondria has remained controversial[2]. Recent evidence indicates that some tumors have rates of glucose oxidation comparable to those observed in normal tissues[3, 4], challenging the assumption that cancer cells are characterized by defective mitochondrial metabolism.

A pathway that relies on functional mitochondria is the oxidation of the third carbon of serine to formate[5]. Formate produced in the mitochondria is released into the cytosol where it supplies the one-carbon demand for nucleotide synthesis[6] (Fig. 1). Formate can also be recycled back to re-synthesize serine via cytosolic one-carbon metabolism[7]. In cells with defective mitochondrial one-carbon metabolism, the cytosolic pathway is reverted compensating for the loss of mitochondrial formate production[7]. When both cytosolic and mitochondrial pathways are compromised cells can utilize exogenous formate[7] or endogenous formaldehyde[8] as alternative sources of one-carbon units.

We predicted[9] and experimentally verified[10] that serine catabolism, and subsequent formate production, often occurs at rates exceeding the one-carbon demand of biosynthesis. The excess formate is then released from the cells, a process referred to as formate overflow. Formate overflow is dependent on the expression of mitochondrial one-carbon metabolism enzymes and competent oxidative phosphorylation[10, 11]. In vitro cell cultures treated with complex I inhibitors[10] or harboring mitochondrial DNA mutations[11] manifest reduced formate release or even switch to formate uptake. Treatment of mice with the complex I inhibitor phenformin inhibits the whole body rate of serine catabolism to plasma formate[10]. However, whether formate overflow is observed in tumors in vivo and if it is dependent on oxidative metabolism remains to be elucidated.

Here we develop experimental methods to investigate the link between formate overflow and redox state in vivo, and to measure the rate of serine catabolism to formate in tissues. Using these methods, we uncover a basal serine catabolism to formate in normal tissues, with tissue specific rates in increasing order of their serine levels and oxidative profile. To estimate the rate of serine catabolism in cancer tissue, we analyzed intestinal adenomas from $APC^{Min/+}$ mice and mammary carcinomas of $PyMT$ mice. Both, the intestinal adenomas and the mammary carcinomas, exhibit significantly increased rates of serine catabolism to formate compared to normal adjacent tissue and other non-tumor-bearing organs. In addition, plasma formate levels were significantly increased in tumor bearing mice compared to wild-type mice in different genetically engineered mouse models (GEMMs) of cancer compared to control mice. This indicates that the tumor-specific high serine catabolism is causative for the elevated plasma formate levels. Finally, we show that inhibition of formate production by genetic knockdown reduces invasion and that this phenotype can be rescued by exogenous formate. We conclude that some cancers are characterized by significant oxidative metabolism, we identify formate overflow as the hallmark of such oxidative cancer types and we propose cell invasion as a possible selective advantage of formate overflow.

## Results

**Formate overflow is controlled by the redox state in vitro**. In vitro studies indicate that formate release requires active mitochondrial oxidative phosphorylation[5, 10, 11]. However, measuring oxidative phosphorylation in vivo is challenging. Since oxidative phosphorylation is a major pathway to oxidize NADH, we reasoned that the $NAD^+/NADH$ redox ratio could be used to investigate the link between the oxidation state and formate release (Fig. 1a). Based on this evidence we formulated the hypothesis that formate release can be used as readout for the whole cell $NAD^+/NADH$ ratio.

To test this hypothesis we analyzed a panel of different cancer cell lines that are routinely used in our laboratory to investigate one-carbon metabolism (Supplementary Table 1). We quantified extracellular formate concentration using a benzyl alcohol derivatization protocol followed by GC-MS[12] and calculated exchange rates of formate release. We also quantified intracellular $NAD^+$ and NADH using LC-MS and calculated the $NAD^+/NADH$ ratio. Plotting the rate of formate release as a function of the $NAD^+/NADH$ ratio uncovered a linear relationship between these two variables across cancer cell lines (Fig. 1b, Pearson Correlation Coefficient PCC = 0.89, p-value = 0.001, excluding the outlier (triangle down)).

To address the validity of our observations in a homogenous genetic background, we analyzed HAP1 cells under different environmental and pharmacological perturbations that alter the cell redox state, albeit by different mechanisms. Here we also introduce the serine catabolism index $SCI = Serine \times NAD^+/NADH$, which takes into account that the serine catabolism rate is determined by both the redox state and the serine concentration[10]. First, we investigated the impact of switching cells from glucose to galactose containing medium, which results in growth inhibition and increases oxidative phosphorylation[10]. When HAP1 cells where cultured in galactose serine levels did not change significantly relative to cells in glucose (Fig. 1c, p = 0.33, unpaired t-test). In contrast, the $NAD^+/NADH$ ratio and consequently SCI increased with a statistical significance close to 0.05 (Fig. 1d (p = 0.06), Fig. 1e (p = 0.08), unpaired t-test). In agreement with the SCI increase, formate release also increased with a statistical significance close to 0.05 (Fig. 1f, p = 0.06, unpaired t-test).

Next, we treated cells with 50 nM of the antifolate methotrexate (MTX), a well-known inhibitor of purine synthesis, thymidylate synthesis and growth (Supplementary Fig. 1). MTX treatment depletes intracellular thymidylate and ATP and suppresses cell proliferation (Supplementary Fig. 1c, e, f), thereby decreasing the rate of NADH generation. Serine levels did not change significantly upon MTX treatment (Fig. 1g, p = 0.44, unpaired t-test). In contrast, $NAD^+/NADH$ and consequently SCI increased (Fig. 1h (p = 0.01), Fig. 1e (p = 0.02) unpaired t-test). The MTX treated cells had a modest increase in oxygen consumption (Supplementary Fig. S1b), indicating that the shift towards more oxidized $NAD^+$ might be a combined effect of increase in oxidative phosphorylation and decrease in growth. More importantly, the MTX dependent shift to a more oxidative state is associated with a significant increase of formate release (Fig. 1j, p = 0.003, unpaired t-test). This observation was surprising given that MTX is an antifolate and one-carbon metabolism is folate dependent. Yet, using [U-$^{13}$C]-Serine tracing we corroborated that there was an increase in $^{13}$C-formate release from cells (Supplementary Fig. 1d). Therefore, at a 50 nM dose, MTX does not inhibit the serine catabolism to formate while growth and one-carbon metabolism dependent nucleotide synthesis is inhibited.

Finally, we tested our hypothesis in the context of hypoxia, where oxygen dependent oxidation processes are inhibited to the degree the oxygen level is reduced. Serine levels did not change significantly in hypoxia (Fig. 1k: p = 0.22 at 1% and p = 0.42 at 0.1%, unpaired t-test). In contrast, $NAD^+/NADH$ and consequently SCI decreased with decreasing oxygen concentrations

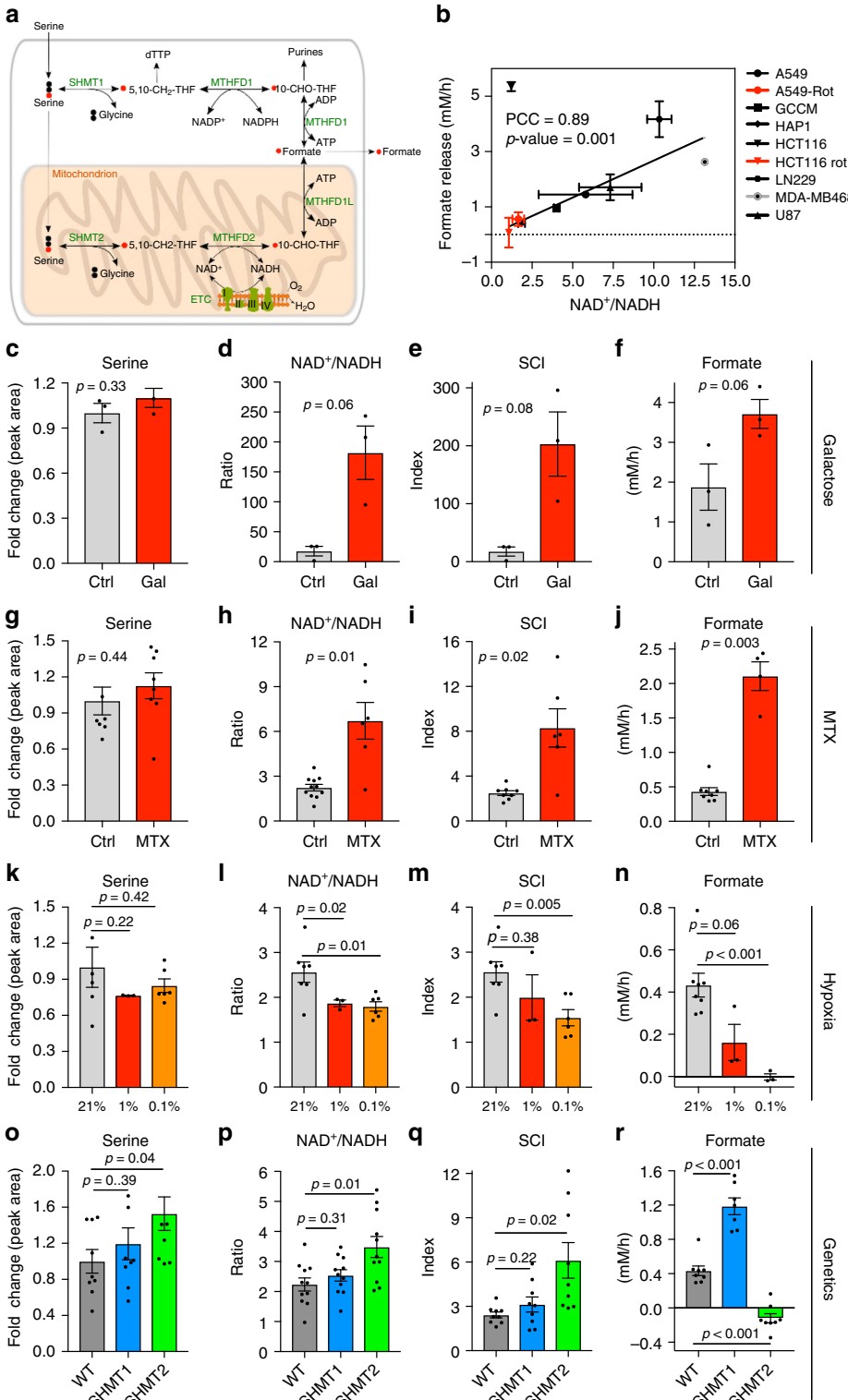

(Fig. 1l: $p = 0.02$ at 1% and $p = 0.01$ at 0.1%; Fig. 1m: $p = 0.38$ at 1% and $p = 0.005$ at 0.1%; unpaired $t$-test). This observation confirms the expectation that hypoxia shifts cells to a more reduced state. In agreement with the SCI decrease, formate release also decreased significantly (Fig. 1n: $p = 0.06$ at 1% and $p < 0.001$ at 0.1%, unpaired $t$-test). Taken together the perturbation analyses support our hypothesis that, regardless of the mechanism, shifting cells towards a more oxidative state induces an increase in formate release, while shifting cells towards a more reductive state decreases formate release.

Previous work[7, 10] indicates that competent mitochondrial one-carbon metabolism is needed to observe formate overflow. However, from that data we cannot exclude that defects in mitochondrial one-carbon metabolism may push cells towards a more reduced state, thereby inhibiting formate overflow. To differentiate between these two factors we used CRISPR-Cas9

engineered HAP1 cell lines that are either wild-type (WT), or defective in cytosolic or mitochondrial one-carbon metabolism by genetic knockout of *SHMT1* or *SHMT2*, respectively[8] (Fig. 1a, Supplementary Fig. 2[8]). The impairment of one-carbon metabolism in *ΔSHMT2* cells is reflected by a significant decrease in their proliferation rate relative to WT or *ΔSHMT1* cells (Supplementary Fig. S2c). *ΔSHMT2* cells exhibit higher serine levels ($p = 0.04$, unpaired $t$-test), higher $NAD^+/NADH$ ratios ($p = 0.01$, unpaired $t$-test) and consequently higher SCI ($p = 0.02$, unpaired $t$-test) than WT or *ΔSHMT1* cells (Fig. 1o–q). Furthermore, *ΔSHMT2* cells have similar basal respiration rates than WT cells (Supplementary Fig. 3). Therefore, the lack of formate release by *ΔSHMT2* cells (Fig. 1r) cannot be explained by a shift towards a more reductive state nor by a decreased activity of oxidative phosphorylation (Supplementary Fig. S3). In fact, despite a similar oxygen consumption rate than WT cells, *ΔSHMT2* cells are in a more oxidative state (higher $NAD^+/NADH$) and they have higher serine levels than WT cells, which should promote one-carbon production from the cytosolic pathway. Indeed, the cytosolic pathway is able to compensate for the one-carbon demand of biosynthesis, as demonstrated by the ~60% $^{13}C$-enrichment in purine one-carbon units in all three HAP1 genotypes (P1C, Supplementary Fig. 4). However, the cytosolic pathway seems to be incapable of reaching the high rates that are needed for formate overflow (Fig. 1r, *ΔSHMT2* cells).

Taken together with our previous data on serine requirement[10], the chemical, environmental and genetic perturbations indicate that serine availability, an oxidative cellular state and competent mitochondrial one-carbon metabolism are necessary conditions to manifest formate overflow.

## $^{13}C$–MeOH protocol to quantify serine catabolism in vivo.

To address the in vivo relevance of these in vitro observations, we developed a $^{13}C$-methanol ($^{13}C$–MeOH) tracing protocol to quantify serine catabolism to formate in vivo (Fig. 2a). Methanol is metabolized in the liver to formate in a two-step reaction: conversion to formaldehyde by alcohol dehydrogenase and subsequent conversion to formate by aldehyde dehydrogenase[13]. Since the liver has a limited capacity for methanol turnover, we hypothesized that an IP bolus of $^{13}C$-methanol may result in a prolonged and steady infusion of $^{13}C$-formate into the blood circulation. To test this hypothesis, an IP bolus of $^{13}C$-methanol (3 g/kg) was injected in wild-type mice and blood and tissue samples were collected at different time points. The $^{13}C$-enrichment in plasma formate, plasma serine, liver serine and spleen serine reached a steady value as soon as one-hour post-injection and was maintained over 24 h (Fig. 2b–e). The $^{13}C$ enrichment of brain serine reached a steady state as well, but at a later time between 3 to 24 h (Fig. 2f). The $^{13}C$-enrichment of purines was detectable at the 20 h time point and reached a steady level between 20 and 24 h (Fig. 2g–i). This slow kinetics was expected given that the purine pool is higher and purine synthesis rates are lower than the corresponding values for serine and formate. Furthermore, different tissues have different rates of purine synthesis that are manifested as different degrees of

$^{13}C$-enrichment in purines (Fig. 2g–i). From the relative abundance of purines with one ($M + 1$) or two ($M + 2$) $^{13}C$-atoms we estimated the $^{13}C$-enrichment of intracellular formate, which should be independent of the rate of purine synthesis. Indeed, the deconvoluted $^{13}C$-enrichment of purines 1C units (P1C) is less variable across tissues than the $^{13}C$-enrichment in purines (Fig. 2j–l). These data indicate that IP injection of $^{13}C$–MeOH can be used to reach isotopic steady state in relevant entities (formate, serine, P1C).

## Baseline rates of serine catabolism to formate in vivo.

Using the $^{13}C$–MeOH tracing protocol we developed an in vivo metabolic flux analysis to determine the relative rate of serine catabolism to formate (SCF) (Fig. 3a). The methodology exploits differences between the $^{13}C$-enrichment in tissue serine, plasma formate and one-carbon units in purines (P1C) (Fig. 2). Serine can be derived from the diet, produced from glucose catabolism or synthesized from glycine and formate. Diet and glucose derived serine dilute the $^{13}C$-enrichment of serine relative to formate (Fig. 2c–f vs. Fig. 2b). Following the same rationale, the one-carbon units in purines can be derived from serine catabolism or from plasma formate (Fig. 3a). Since the $^{13}C$-enrichment of tissue serine is always lower than the $^{13}C$-enrichment of plasma formate (Fig. 2d–f vs. Fig. 2b), the observation that the $^{13}C$-enrichment in P1C is lower than the $^{13}C$-enrichment of plasma formate is evidence that there is catabolism of serine to formate. The larger is the difference, the larger is the relative contribution of serine catabolism to the one-carbon pool of the tissue.

Applying this methodology, we estimated the baseline rate of serine catabolism to formate in the context of normal physiology and its relation with the tissue redox state. We profiled brain, liver and spleen tissues that are characterized by distinctive metabolic features. We focused on the two major factors affecting serine catabolism in vitro, serine levels and the $NAD^+/NADH$ ratio. Serine levels were the highest in the brain, intermediate in spleen, and lowest in the liver (Fig. 3b). The $NAD^+/NADH$ ratio was the highest in the spleen, intermediate in the brain and lowest in liver (Fig. 3c). When combined together to calculate the SCI, brain exhibits the highest SCI, spleen intermediate and liver the lowest (Fig. 3d). This pattern is confirmed by the relative rate of serine catabolism to formate (SCF, Fig. 3e). SCF also exhibits its highest value in the brain, intermediate in the spleen and lowest in the liver tissue. Using the $^{13}C$–MeOH tracing protocol we have also estimated the purine fraction that has been de novo synthesized within the 24 h time window (Fig. 3f). The fraction of de novo synthesized purines was higher in the liver (Fig. 3f), where the estimated serine catabolism to formate is close to zero (SCF, Fig. 3e). Furthermore, the fraction of de novo synthesized purines was higher in the spleen than in the brain, the opposite of what we determined for the relative rate of serine catabolism to formate (SCF, Fig. 3e). Taken together these data indicate that in vivo serine catabolism to formate is associated with serine levels and the tissue redox state, but is not correlated with the rate of purine synthesis.

---

**Fig. 1** Formate overflow is controlled by the cell redox state in vitro. **a** Diagram of mammalian one-carbon metabolism with serine as the major substrate for both anabolic and catabolic processes. **b** Correlation of formate release with the cell redox state estimated by the $NAD^+/NADH$ ratio. Symbols indicate different cell lines (see legend and Supplementary Table 1). Black symbols indicate standard cell culture conditions, red symbols indicate treatment with the complex I inhibitor rotenone (250 nM). The line represents a linear fit excluding outlier (triangle down). **c–f** Metabolic profile of HAP1 cells growing in regular medium or medium with galactose (Gal) instead of glucose (Ctrl). **g–j** Metabolic profile of HAP1 untreated cells (Ctrl) and cells treated with 50 nM MTX. Significant growth repression upon MTX is illustrated in Supplementary Fig. S1c. **k–n** Metabolic profile of HAP1 cells under ambient oxygen (21%) or moderate (1%) and deep (0.1%) hypoxia. **o–r** Metabolic profile of HAP1 parental, *ΔSHMT1* and *ΔSHMT2* cells. For knockout validation and growth analysis see Supplementary Fig. S2. Each dot indicates one independent experiment performed with triplicate wells. Each experiment was performed at least three times. Error bars indicate s.e.m. *p*-values are calculated using an unpaired *t*-test with Welch's correction. PCC Pearsson correlation coefficient

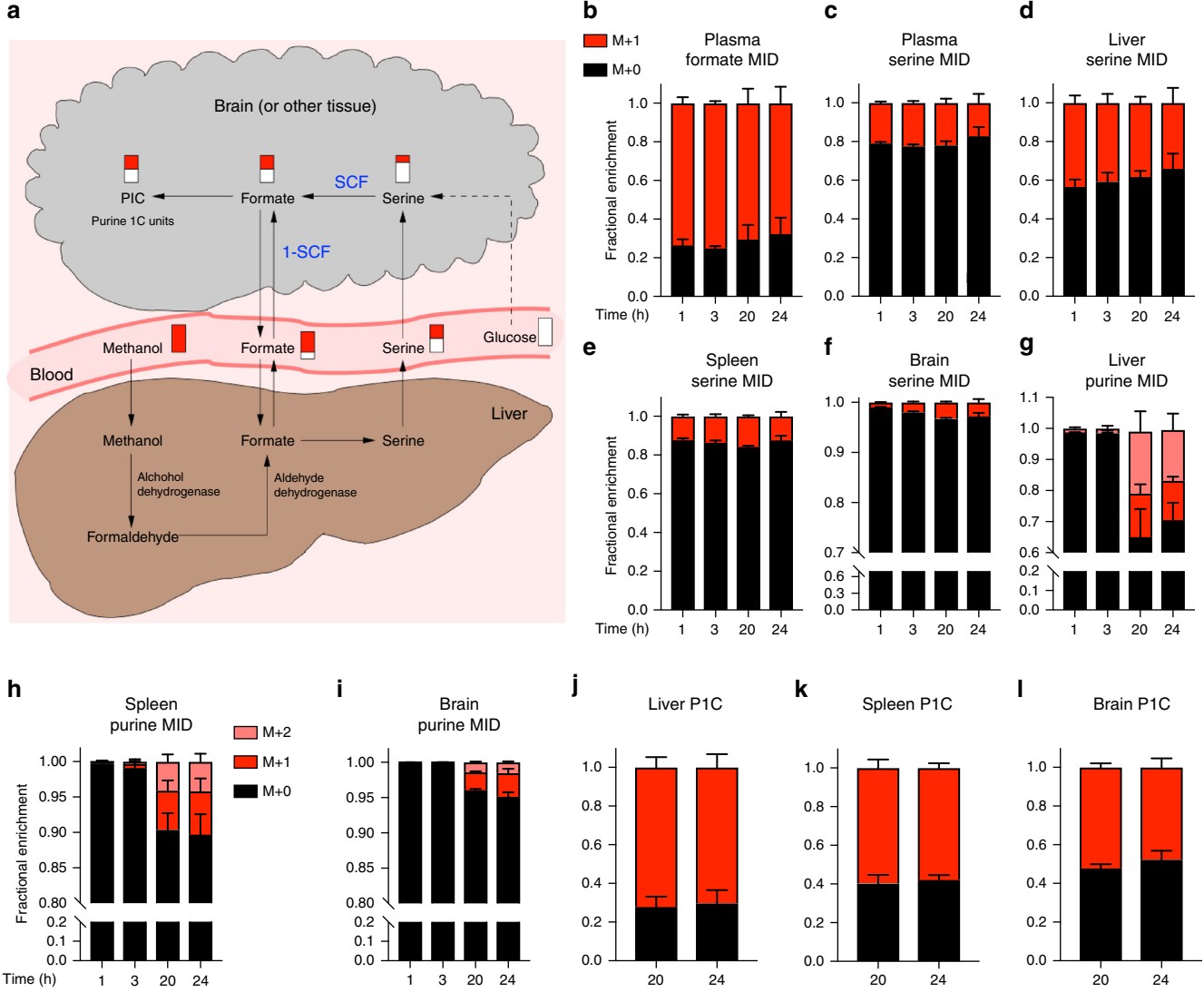

**Fig. 2** $^{13}$C–MeOH protocol to quantify serine catabolism to formate in vivo. **a** Diagram tracing the metabolism of $^{13}$C methanol and its intersection with serine catabolism to formate. **b**–**i** time-dependent Mass Isotopomer Distribution (MID) of different metabolites in different tissues after a 3 g/kg IP injection of $^{13}$C–MeOH. **b**, **c** Plasma formate and serine MID. **d**–**f** Serine MID in liver, spleen and brain. **g**–**i** Purine MID in liver, spleen and brain. **j**–**l** P1C MID in total purines in liver, spleen and brain. Results were obtained from four mice per time point ($n = 4$). Error bars denote s.e.m

**Increased serine catabolism to formate in APC$^{Min/+}$ adenomas.**
To determine whether serine catabolism to formate is increased in cancer tissue we investigated the $APC^{Min/+}$ model of intestinal cancer. We analyzed tissue samples from brain, liver, spleen, normal small intestine (N-SI) and adenoma bearing small intestine (A-SI) of $APC^{Min/+}$ mice with clinical manifestation of the disease (Fig. 4a–d). Serine levels were found the highest in adenomas of the small intestine, relative to adjacent non-transformed small intestine tissue ($p = 0.008$, unpaired $t$-test) and other normal tissues as well (Fig. 4a). In adenomas, the NAD$^+$/NADH ratio was lower than the highest values observed in spleen ($p = 0.03$, unpaired $t$-test), but significantly higher than in the adjacent non-transformed small intestine tissue ($p = 0.001$, unpaired $t$-test) (Fig. 4b). When combined together to calculate the SCI, the adenomas exhibit the highest value, significantly higher than in the adjacent non-transformed small intestine tissue ($p = 0.004$, unpaired $t$-test) (Fig. 4c). These findings indicate a high tumor-specific potential for serine catabolism to formate. We note that the increase of SCI in the intestinal adenomas is due to both an increase in serine levels and the NAD$^+$/NADH ratio.

Next, we deployed the $^{13}$C–MeOH tracing protocol to quantify the rate of serine catabolism to formate and to test whether it is indeed increased in the adenomas of $APC^{Min/+}$ mice relative to normal adjacent tissue. As observed in wild-type mice (Fig. 3), the $^{13}$C from MeOH was incorporated into formate, serine and purines in all tissues profiled of $APC^{Min/+}$ (Supplementary Fig. 5a–d). From the measured $^{13}$C-enrichment in plasma formate, tissue serine and the one-carbon of purines (Supplementary Fig. 5a–d), we estimated the relative rate of serine catabolism to formate (Fig. 4d). As reported above for wild-type mice (Fig. 3e), the brain and spleen of $APC^{Min/+}$ exhibit a significant rate of serine catabolism to formate. More importantly, the rate of serine catabolism to formate in adenomas is significantly higher than in the small intestine tissue adjacent to the adenomas ($p < 0.001$, unpaired $t$-test) and all the other tissues analyzed (compared to brain: $p = 0.02$, compared to spleen: $p = 0.01$, unpaired $t$-test).

**Increased serine catabolism to formate in PyMT tumors.** As a second independent model, we have investigated the PyMT

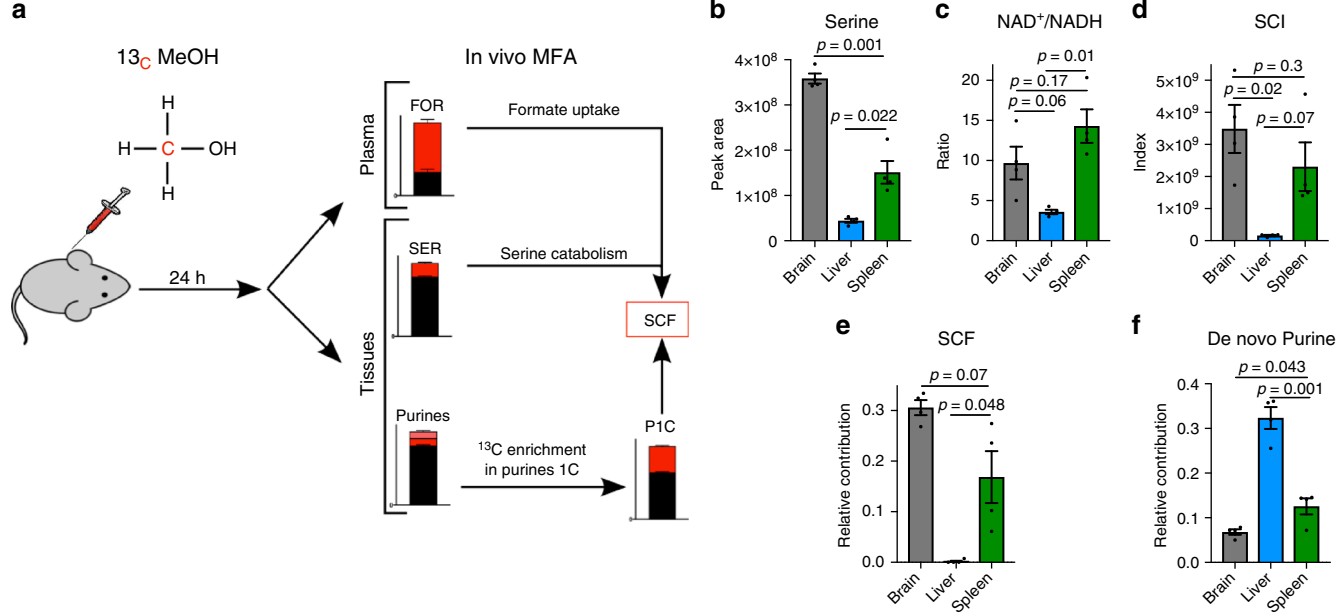

**Fig. 3** Baseline rates of serine catabolism to formate in murine tissues. **a** Diagram illustrating the $^{13}$C-methanol tracing and in vivo metabolic flux analysis protocol. **b–d** Serine, NAD$^+$/NADH and SCI for brain, liver and spleen samples from mice 24 h post $^{13}$C–MeOH injection. **e** Relative rate of serine catabolism to formate (SCF) of brain, liver and spleen as estimated from the $^{13}$C–MeOH tracing protocol. **f** De novo purine synthesis rate in brain, liver and spleen inferred from the $^{13}$C enrichment of purines. Results were obtained from three mice ($n = 3$). Error bars denote s.e.m.. $p$-values are calculated using an unpaired $t$-test with Welch's correction

model for mammary cancer. We analyzed tissue samples from brain, liver, spleen, normal mammary gland (N-MG) and mammary gland tumors (T-MG) of PyMT mice with clinical manifestation of the disease (Fig. 4e–h). The data is strikingly similar to that for the $APC^{Min/+}$ model. Serine levels were found the highest in the tumors, relative to adjacent non-transformed mammary gland ($p = 0.048$, unpaired $t$-test) and other normal tissues except brain (Fig. 4e). NADH levels where below detection in some normal and in some tumor samples (Supplementary Fig. S6). This indicates that mammary tissue is quite oxidative and that this oxidative state is preserved in the tumor. To overcome the caveat of dealing with zero or very low values of NADH, we replaced the NAD$^+$/NADH by the NAD$^+$/(NADH + NAD$^+$) ratio. The latter is less sensitive to redox variations but it can cope with scenarios where NADH is zero. The NAD$^+$/(NADH + NAD$^+$) ratio in mammary tumors was comparable to the highest values observed in spleen and normal mammary gland (Fig. 4f). When combined together to calculate the SCI* (here defined as Serine x NAD$^+$/(NADH + NAD$^+$)), the brain tissue and the mammary tumors exhibit the highest values (Fig. 4g). We note that the increase of SCI in the mammary tumors compared to adjacent non-transformed mammary gland tissue ($p = 0.043$, unpaired $t$-test) is mainly due to an increase in serine.

Next, we deployed $^{13}$C–MeOH tracing to quantify the rate of serine catabolism to formate in the different tissues in PyMT mice. Here again the $^{13}$C from methanol was incorporated into formate, serine and purines in all tissues profiled of PyMT (Supplementary Fig. 5e–h), the data that is utilized to estimate the relative rate of serine catabolism to formate (Fig. 4h). As reported above for wild-type mice (Fig. 3e) and end-stage $APC^{Min/+}$ mice (Fig. 4d), the brain and spleen of end-stage PyMT mice exhibit a significant rate of serine catabolism to formate. More importantly, the rate of serine catabolism to formate in the mammary tumors is significantly higher than in the mammary gland tissue adjacent to the tumors (Fig. 4h, $p = 0.002$, unpaired $t$-test). The rate of serine catabolism to formate in the mammary tumors is also

significantly higher than in the brain (Fig. 4h, $p = 0.02$, unpaired $t$-test), the tissue with highest value among all non-transformed tissues analyzed.

**Increased plasma formate levels in GEMMs**. We have put together all our estimates of serine catabolism to formate in wild-type, $APC^{Min/+}$ and PyMT mice (Fig. 4i). This aggregate data shows that all normal tissues profiled, have no significant changes between wild-type and tumor bearing mice in the relative rate of serine catabolism to formate. There is a trend towards increased serine catabolism to formate in the spleen of $APC^{Min/+}$ mice with clinical manifestation of the disease ($p = 0.08$, unpaired $t$-test) (Fig. 4i). However, there is a significant and more dramatic increase of the relative rate of serine catabolism to formate in the adenoma/tumor tissues relative to adjacent normal tissue (Fig. 4i: $p < 0.001$ for small intestine and $p = 0.002$ for mammary gland, unpaired $t$-test). Overall, this data indicates a tumor-specific increase in the relative rate of serine catabolism to formate.

To determine whether the tumor-specific increase in serine catabolism to formate results in increased formate levels in circulation, we have quantified plasma formate levels in wild-type and tumor-bearing mice. First, we analyzed the serum of $APC^{Min/+}$ mice (80 days old pre-neoplastic and endpoint) and wild-type controls. 80 days aged $APC^{Min/+}$ mice show a significant increase in serum formate levels compared to controls ($p = 0.005$, unpaired $t$-test) (Fig. 5a). Even though with less significance, increased formate levels were sustained until the mice reached clinical endpoint ($p = 0.09$, unpaired $t$-test) (Fig. 5a). Since serine catabolism to formate was increased in small intestinal adenomas (Fig. 4d), this data indicates that the increased serum formate levels reflect the increased serine catabolism in the adenomas. Similar evidence was obtained when analyzing serum samples from the PyMT model of mammary cancer. Tumor bearing PyMT mice show a significant increase in serum formate levels compared to controls ($p = 0.001$, unpaired $t$-test) (Fig. 5b).

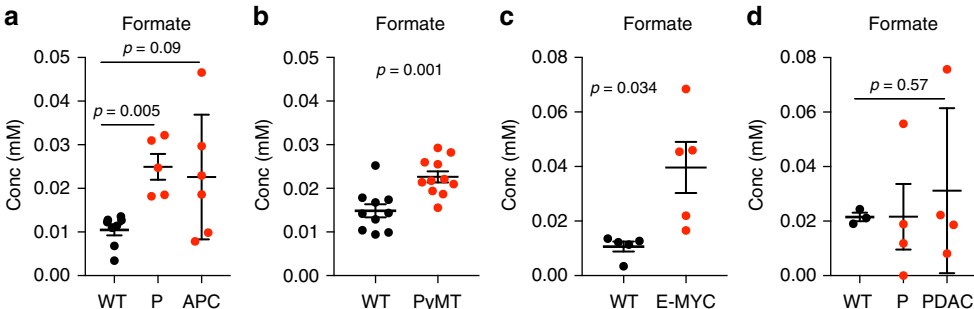

**Fig. 4** Increased serine catabolism to formate in $APC^{Min/+}$ adenomas and PyMT tumors. **a–d** Metabolic analysis of $APC^{Min/+}$ mice with clinical manifestation of the disease. **a** Serine levels, **b** NAD⁺/NADH ratio **c** SCI and **d** SCF in brain, liver, spleen, normal small intestine (N-SI), and in small intestine adenomas (A-SI) 24 h after $^{13}C$–MeOH injection. **e–h** Metabolic analysis in PyMT tumor-bearing mice (as in **a–d**). Except for **f**: here the NAD⁺/(NADH + NAD⁺) ratio was assessed due to technical reasons. **g** *Accordingly, in this case the SCI was calculated by serine × NAD⁺/(NADH + NAD⁺). MG mammary gland, T tumor, N normal. **i** Comparison of SCF in liver, spleen and brain in WT, $APC^{Min/+}$, and PyMT mice and SI and MG tissues. Results were obtained from at least three mice. Error bars denote s.e.m. p-values are calculated using an unpaired t-test with Welch's correction

**Fig. 5** Increased serum formate levels in GEMMs. **a–d** Serum formate in wild-type (WT), pre-neoplastic (P) and neoplastic tissues from different GEMMs for **a** intestinal cancer, **b** mammary, **c** lymphoma and **d** pancreatic ductal adenocarcinoma (PDAC). Results were obtained from at least four mice. Error bars denote s.e.m. p-values are calculated using an unpaired t-test with Welch's correction

Finally, we quantified formate levels in two other GEMMs for cancer: lymphoma [Eµ-Myc] and pancreatic cancer [KPC]. We found a significant increase in serum formate in the serum of Eµ-Myc endpoint mice ($p = 0.034$, unpaired $t$-test), but not in the serum of KPC endpoint mice ($p = 0.57$, unpaired $t$-test) (Fig. 5c,d).

Therefore, our findings indicate that the increased rate of serine catabolism to formate in adenomas/carcinomas can have a significant impact on peripheral levels of formate. It is interesting to note that not all tumor-bearing mice have elevated formate levels relative to wild-type controls. We have already identified the KPC model as a counterexample.

**Formate promotes cancer cell invasion in vitro.** Increased formate overflow could be a byproduct of increased serine catabolism to sustain tumor growth. However, as reported above for normal tissue physiology, there is no association between the relative rate of serine catabolism to formate and purine synthesis (Fig. 3e, f). An alternative hypothesis is that formate overflow confers tumor cells with a selective advantage, independent of growth. Since invasion is a hallmark of cancer associated with the remodeling of the tumor microenvironment[14], we investigated whether formate levels modulate invasion. Towards this end, we selected glioblastoma multiforme (GBM) as a tumor type representative for a high degree of invasiveness. We used the GBM cell lines U87 and LN229 out of our profiled cell line panel (Fig. 1b). We also included NCH601 cells in our analyses, because of their reported level of invasiveness[15].

We performed invasion assays with these glioma cells using Boyden chambers (Fig. 6a) to measure their invasion potential. First, we titrated sodium formate in the cell culture medium of wild type cells. Formate supplementation increased cell invasion in a concentration dependent manner in all three cell lines tested (Fig. 6b–d and Supplementary Fig. 9), indicating that formate promotes glioma cell invasion. Next, we generated two independent stable shRNA knockdown cell lines for *MTHFD1L*, the gene encoding for the mitochondrial enzyme responsible for formate production (Fig. 1a). The knockdown was carried out in LN229 cells and NCH601 cells (shMTHFD1L-1 and -2), along with non-targeting shRNA controls (shCtrl). The knockdown was validated by a decrease in protein expression and a reduction in formate release (Supplementary Figs. 7 and 8). Except for one clone of LN229 cells (LN229-shMTHFD1L-2), stable *MTHFD1L* knockdown resulted in a significant reduction of invasion relative to shCtrl (Fig. 6e, f and Supplementary Figs. 10 and 11), providing genetic evidence that endogenous serine one-carbon catabolism with formate overflow promotes invasion. Finally, addition of exogenous formate rescued the invasion reduction in the shMTHFD1L knockdown cells (Fig. 6e, f and Supplementary Figs. 10 and 11). To exclude growth effects that could impact the number of invaded cells, we monitored growth of shMTHFD1L cell lines versus shCtrl cells including different formate concentrations. All cell lines grew at similar rates independent of the genetic or chemical perturbation (Supplementary Fig. 7). Furthermore, we validated that addition of

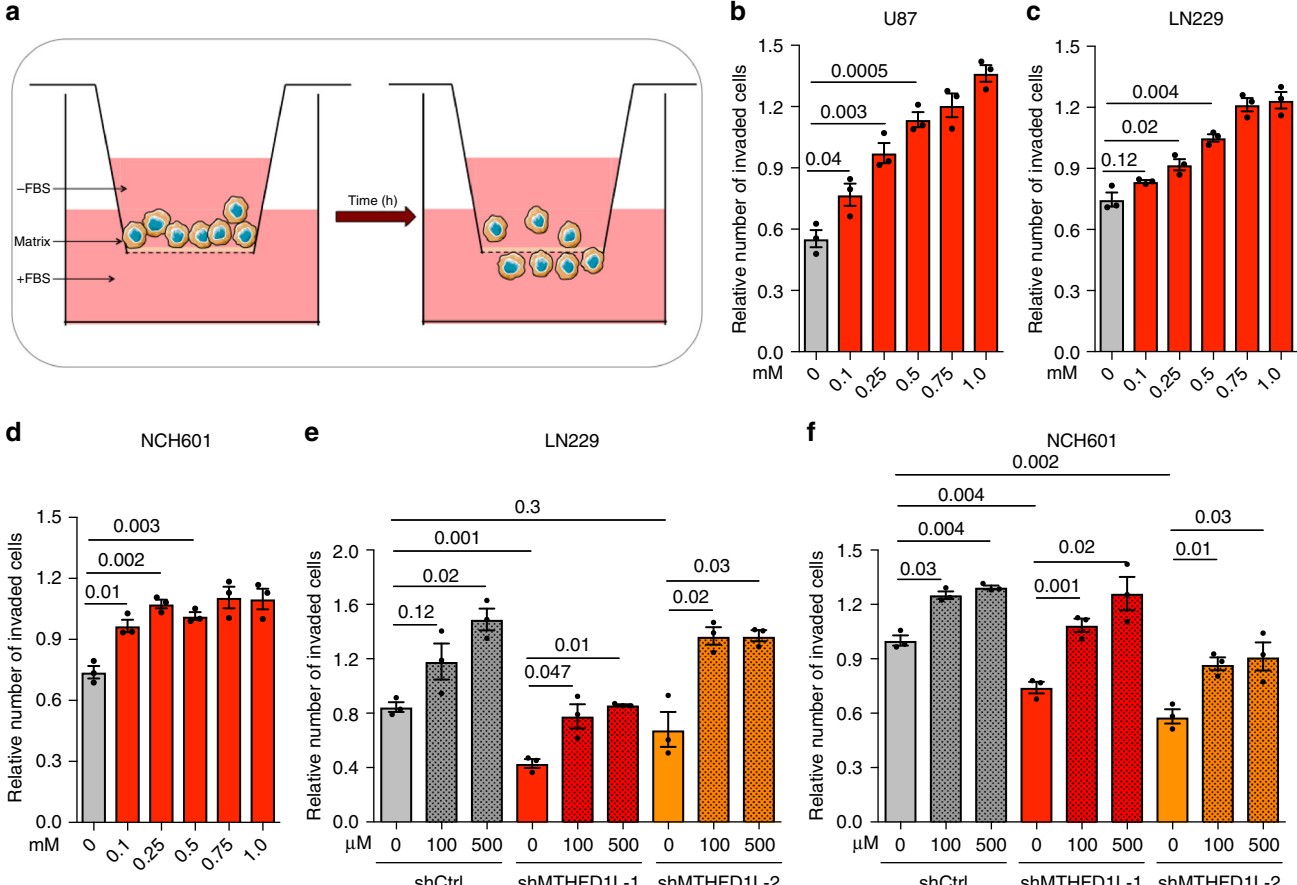

**Fig. 6** Formate overflow promotes cancer cell invasion in GBM. **a** Cartoon illustrating the experimental setup to analyze cancer cell invasion using coated Boyden chambers. **b**–**d** Addition of extracellular sodium formate increases invasiveness in a concentration dependent manner in **b** U87, **c** LN229 and, **d** NCH601 cells. **e**, **f** Reduced invasiveness by *MTHFD1L* knockdown can be rescued with extracellular formate in **e** LN229 and **f** NCH601 cells. Each dot indicates one independent experiment ($n = 3$). Error bars indicate s.e.m. $p$-values are calculated using an unpaired $t$-test with Welch's correction

sodium-formate to the growth medium did not alter the pH. Taken together these data indicate that high levels of formate generated by mitochondrial one-carbon metabolism or from extracellular sources promote glioma cell invasion, either by an intrinsic mechanism or by increasing extracellular formate concentration.

## Discussion

The in vitro evidence reported here, combined with previous work[5, 10, 11], demonstrates that serine availability, an oxidative cellular state, and a competent mitochondrial one-carbon metabolism are necessary conditions to manifest formate overflow. The analysis of in vivo models of cancer indicates that both increased serine levels and a highly oxidative state, can be manifested in transformed tissues. These transformed tissues represent an environment promoting serine catabolism to formate. Indeed, we determined that the relative rate of serine catabolism to formate is increased in the analyzed transformed tissues relative to adjacent normal tissues and is the highest among all tissues analyzed. This tumor-specific increase of serine catabolism to formate is associated with increased levels of circulating formate. Compared to normal controls, plasma levels of formate were increased in GEMMs for intestinal adenoma, breast cancer and lymphoma, but not for pancreatic cancer indicating that formate overflow can be a frequent phenotype in tumors[16].

Based on this evidence we propose an update of the key features defining cancer metabolism. There is a category of cancers with putative poor oxidative metabolism, that are formate overflow negative and most likely highly glycolytic (Fig. 7a). That is cancer metabolism as envisioned by Warburg[1]. Yet, there is another category of cancers with putative active oxidative metabolism that are formate overflow positive (and potentially highly glycolytic as well, Fig. 7b). The oxidative nature of such cancers is so prominent that formate overflow translates to increased formate concentrations in the circulation.

Our data also provide new evidence about a potential selective advantage of increased mitochondrial one-carbon metabolism for cancer development. It is well established that mitochondrial one-carbon catabolism supplies the one-carbon units for cancer growth[6, 17, 18]. Yet, in the absence of mitochondrial serine catabolism, the cytosolic pathway can sustain cancer growth both in vitro and in vivo[7]. Moreover, in the absence of both cytosolic and mitochondrial serine catabolism, other sources of one carbon units such as endogenous and exogenous formaldehyde can sustain growth[8]. Based on the work presented here, we identified an additional selective advantage, that high rates of mitochondrial serine catabolism to formate, in excess of the biosynthetic demand, promote invasion. We emphasize that

formate production needs to be in excess of biosynthetic demands because knockdown of *MTHFD1L* does not inhibit proliferation but, knockdown of *MTHFD1L* inhibits formate overflow and the cell's invasiveness, indicating that excess formate production is required to promote the invasion phenotype. The mechanism of how formate promotes invasion remains to be elucidated. Excess formate production or formate supplementation may increase intracellular formate levels and high intracellular formate could be the signal promoting invasion. There is also the possibility of non-cell autonomous effects, whereby formate overflow from a group of cells could promote invasion by nearby cells. Additional research is required to unravel the underlying mechanism on how formate promotes invasion.

There have been previous reports indicating that knockdown of *MTHFD2*, encoding for the enzyme catalyzing the second step of mitochondrial one-carbon metabolism, reduces invasion[19, 20]. However, this effect could have been mediated by one or more of the different products of mitochondrial one-carbon metabolism (Fig. 1a, formate, glycine, NAD(P)H and ATP). What is novel about our observations is that the promotion of invasion is mediated by formate. Addition of exogenous formate rescues the inhibition of mitochondrial one-carbon metabolism. This link between formate and invasion may also be relevant in the context of embryonic development. Homozygous deletion of genes encoding for mitochondrial serine catabolism enzymes is embryonic lethal in mice[21, 22] and the embryonic lethality can be rescued by formate supplementation[22]. The parallel between these observations in the context of embryonic development and our observations in the context of cancer invasion assays suggest a common explanation. It is worth mentioning that the major phenotype associated with the embryonic lethality is neural tube defects. Further research is required to determine whether a reduction in formate overflow is the cause of the neural tube defects.

Our findings open a whole array of new questions. Further studies are needed to validate these results in a patient cohort, study the implications of increased formate levels in the tumor and blood and its potential as a biomarker for certain cancers. At present, it is not clear which molecular, genetic or environmental factors determine whether a tumor will exhibit formate overflow and how the non-oxidative *versus* oxidative nature of cancers will affect their response to cancer therapy. Finally, the mechanism of how formate promotes invasion remains to be elucidated.

## Methods

**Chemicals**. Methotrexate (sigma; M8407) was applied at a final concentration of 50 nM for 16–20 h. Rotenone (sigma; R8875) was applied at a final concentration of 250 nM for 16–24 h. Galactose (sigma; G0750) was used instead of glucose at 17.5 mM in DMEM 5030 (thermo, A14430-01). Na-Formate was purchased from Sigma (Nr. 71539).

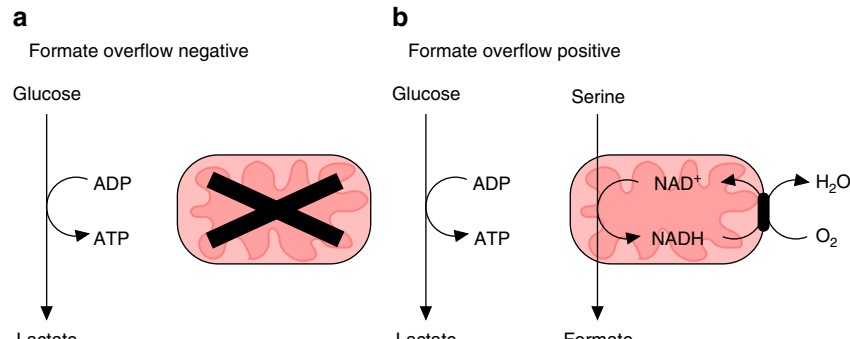

**Fig. 7** Two putative metabolic classes of cancers. **a** Cancers with low oxidative metabolism. These cancers are expected to be formate overflow negative. **b** Cancers with active oxidative metabolism. These cancers are expected to be formate overflow positive

**Culturing and metabolic characterization of cell lines**. All cell lines listed in Supplementary Table 1 were obtained from American Type Culture Collection (ATCC). Cells were regularly (once per month) tested for mycoplasma contamination. No contaminations have been observed within the performed experiments. Cells were cultured in DMEM 5030 supplemented with 17.5 mM glucose, 2 mM glutamine and 10% FBS at 5% $CO_2$ and 37 C. HAP1 cells were obtained from Patel's laboratory[8] and were cultured in IMDM medium supplemented with 10% FBS. NCH601 cells (derived from the lab of Christel Herold-Mende, Heidelberg) were cultured as non-adherent spheres in DMEM-F12 medium (Lonza) containing 1xBIT100 (Provitro), 2 mM glutamine, 30 U/ml Pen/Strep, 1 U/ml heparin (Sigma), 20 ng/ml bFGF (Miltenyi) and 20 ng/ml EGF (Provitro) as previously described in[15, 23].

For metabolic profiling we deployed previously established protocols for absolute quantification of exchange fluxes and intracellular fluxes of one-carbon metabolism[10, 12, 24] to a panel of cancer cell lines. To determine the growth rate, we assumed exponential growth and recorded start count and end count in every experiment and calculated the growth rate as described in[10]. For the determination of the SCI we calculated the product of serine level multiplied with the NAD$^+$/NADH ratio.

**shRNA-mediated gene silencing**. Gene knockdown with shRNA was performed in LN229 and NCH601 cells using lentiviral particles expressing two different shRNAs targeting the *MTHFD1L* gene and one non-targeting shRNA (shCtrl: TTACTCTCGCCCAAGCGAG (Code: RHS4346); shMTHFD1L-1: ATGTGAGT-GATGTTCAGAC (Code: V2LHS_96541); shMTHFD1L-2: TAGATTTCAATTT-CATCTG (Code: V2LHS_96542)). Plasmids containing shRNA sequences were ordered from Dharmacon. Lentiviral particles were produced in HEK293T cells by co-transfection of the pGIPZ-shRNA-control or pGIPZ-shRNA-target gene vector with the viral core packaging construct pCMVR8.74 and the VSV-G envelope protein vector pMD2.G. Supernatant containing viral particles was used to transduce $10^6$ (LN229) or $4 \times 10^5$ cells (NCH601) and puromycin selection (1 µg/ml) was applied to obtain stably transduced GFP-positive cells.

**Western blot**. Protein extracts were resolved in Novex 4–12% BisTris gels (Life Technologies), and blotted onto a nitrocellulose membrane (Life Technology) according to standard protocols and blocked with milk powder. Blots were incubated at 4 °C overnight with primary antibodies specific for MTHFD1L (16113-1-AP, Proteintech, diluted 1:1000), LaminB1 (ab16048, Abcam, diluted 1:1000), SHMT1 (HPA023314, Sigma, diluted 1:250), SHMT2 (HPA020543, Sigma, diluted 1:250) or Vinculin (SAB4200080, Sigma, diluted 1:100000). Secondary antibodies for the detection of primary antibodies against MTHFD1L and LaminB1 were coupled to Horseradish peroxidase (HRP) (Goat anti rabbit HRP; 111-036-003; Jackson ImmunoResearch, diluted 1:10000). To detect primary antibodies against SHMT1, SHMT2 and Vinculin IRDye® 800CW Donkey anti-Rabbit IgG (H + L) (for SHMT1 and 2) or –anti mouse (for vinculin) was used (925–32213 (anti rabbit, diluted 1:10,000), 925–32212 (anti mouse, diluted 1:10000), LiCor). Secondary antibodies were incubated for 1 h at RT. In the case of HRP, signals were detected with a chemiluminescent substrate (ThermoScientific) and imaged using the ImageQuant 350 scanning system (GE Healthcare). In the case of IRDye, signals were detected with a LiCor Odyssey image analyzer. For image analysis ImageStudioLite Software was used.

Western Blots are presented in Supplementary Figs. 2 and 7. Uncropped scans (including a molecular weight marker) are presented in Supplementary Figs. 2 and 8.

**Invasion assays**. The in vitro invasion potential was determined on collagen/ECM-coated (0.05 mg/ml collagen type I (Sigma-Aldrich), 0.5 mg/ml protein of ECM gel) transwell chambers with 8 µm pore size (Greiner Bio-one). For this, 300 µl cell suspension without FBS (with $5 \times 10^4$ cells) was added into the transwell in duplicates per condition. 10% FBS was added to the medium in the bottom well (750 µl total volume) as chemoattractant. In case of formate supplementation Na-Formate (Sigma) was spiked from 100× stocks into the chambers. After 18 h (LN229 and U87) or 72 h (NCH601), cells were fixed with 4% PFA and stained with 0.05% crystal violet solution for 15 min, respectively. Non-invading cells were removed at the top of the chambers with Qtips and the invasion of GBM cells was evaluated by counting the cells on the lower side of the membrane using a light microscope at ×20 magnification (5 view fields/chamber).

**Growth assay**. For growth assay of LN229 and NCH601 cells the IncuCyte instrument was used. For LN229 cells, $2.5 \times 10^3$ cells were seeded in 96-well plates and imaged every 3 h using IncuCyte (Essen BioScience) (brightfield, ×10 objective, 66 h). Confluence was determined using the Incucyte analysis software. To evaluate the NCH601 sphere growth over 93 h, $1 \times 10^3$ cells were plated in 384 wells. The plate was shortly centrifuged before the experiment start to center the cells. Plates were imaged and analyzed similar to LN229.

For growth analysis of HAP1 cells, 50,000 cells were plated in 12 well plates. Growth was monitored over three days with one count per day.

**Measurement of oxygen consumption rate (OCR)**. OCR was measured using the Extracellular Flux analyzer (Seahorse Biosciences) as described in ref.[10]. For MTX treatment: medium was replaced right before the assay with medium containing 50 nM MTX.

**Animal models**. All in vivo experiments were carried out in dedicated barriered facilities proactive in environmental enrichment under the EU Directive 2010 and Animal (Scientific Procedures) Act (HO licence numbers: 70/8645, 70/8468 and 70/8375) with ethical review approval (University of Glasgow). Animals were cared for by trained and licensed individuals and humanely sacrificed using Schedule 1 methods.

*Wild-type* male C57BL/6J mice (stock number, 000664; at least 20 g in weight) were purchased from Charles River.

*APC^Min/+* mice[25] were bred in house on a C57BL/6J background (N6 or above). Mice were monitored at least 2 times weekly and culled at 80 days or when exhibiting symptoms of intestinal adenoma burden (weight loss, paling feet, hunching). Tissues (serum, spleen, liver, and brain) were harvested and flash frozen on dry ice. The small intestine was flushed with PBS; normal small intestinal tissue and tumor tissues were dissected and snap frozen on dry ice.

Mouse Mammary Tumor Virus (MMTV) Polyoma Middle-T antigen (PyMT)[26] (FVB/N >20 generations) female mice (hereafter PyMT mice) were killed humanely when mammary tumors reached clinical endpoint as per regulation. Litter-matched wild-type FVB females were used as controls.

*Eu-MYC* mice[27] (>N20 C57BL/6J) were bred in house and monitored twice weekly until manifestation of disease onset. Lymphomatous tissues (spleen/lymph node) were harvested and flash frozen on dry ice.

KPC (*Pdx1-Cre; LSL-Kras^G12D/+; LSL-Trp53^R172H*)[28] mice were bred in house on a mixed background. Mice were monitored at least 3 times weekly and culled when exhibiting symptoms of pancreatic ductal adenocarcinoma (PDAC, swollen abdomen, loss of body conditioning, jaundice, immobility or hunching), or before the onset of symptoms or palpable tumor for pre-neoplastic lesion-enriched tissue. Tissue was flash frozen immediately upon harvest.

**Tissue analysis**. For each tissue one sample was generated per mouse, except for small intestine and adenomas of small intestine. Here, 3–4 samples were analyzed per mouse. Flash frozen tissue samples were blinded with random IDs and processed by a different person for metabolite extraction and analysis. After final data analysis IDs were uncovered. Replicate values per mouse were pooled and the mean value was used for further data analysis. All tissues were processed frozen on dry ice. From each tissue 5–20 mg were balanced and transferred into Precellys CK14 tubes (Bertin Technologies, Montigny-le-Bretonnex, France). Tissues were dissolved in 20 mg/ml extraction solvent (Acetonitrile/MeOH/ $H_2O$ (30/50/20)) and homogenized in a cooled Precellys 24 (Bertin Technologies, Montigny-le-Bretonnex, France) with $3 \times 20$ s at 7200 r.p.m. and a 20 s break. Lysed tissue samples were transferred into Eppendorf tubes and centrifuged for 10 min at 4 °C. Supernatant was transferred into LC-MS vials for mass spec analysis.

**13C-Methanol tracing**. For $^{13}$C–MeOH tracing in wild type mice, mice were randomized in individual groups (different time points and PBS controls). Mice were sacrificed at respective time points. For $^{13}$C–MeOH tracing in APC^Min/+ and *PyMT* mice, mice were injected intraperitoneally with 3 g/kg $^{13}$C–MeOH (Eurisotop; CLM-359-1) when reaching clinical endpoint and sacrificed 24 h post injection. Blood was immediately taken by cardiac puncture, transferred into Eppendorf tubes and centrifuged at 4 °C for 10 min at 13k G. The supernatant was transferred into new Eppendorf tubes and flash frozen in liquid nitrogen. Tissues were harvested in Eppendorf tubes, flash frozen in liquid nitrogen and analyzed as described above. All in vivo samples were blinded with random IDs and processed by a different person for metabolite extraction and analysis. After final data analysis IDs were uncovered.

**Metabolite extraction and analysis**. Metabolite extraction and analysis was performed as previously described[10, 29]. Briefly, cells were washed with PBS once and extracted with ice-cold extraction solvent (Acetonitrile/MeOH/ $H_2O$ (30/50/20)), shaken for 5 min at 4 °C, transferred into eppendorf tubes and centrifuged for 5 min at 18k G. The supernatant was transferred in LC-MS glass vials and kept at −80 °C until measurement. Formate extraction and derivatization was performed as described in[10] using a Methylchloroformate derivatization approach. Derivatized formate was analyzed using GC-MS (Agilent). Heavy labeled M + 2 formate was used as internal standard for quantification.

LC-MS analysis was performed as described previously[10] using HILIC chromatography and a Q-Exactive mass spectrometer (Thermo Fisher Scientific). Raw data analysis was performed using TraceFinder (Thermo Fisher Scientific) software. Peak areas were normalized to cell volume in case of in vitro experiments, or to wet weight (mg) tissue.

**13C enrichment in purine one carbon units (P1C)**. To calculate the $^{13}$C enrichment in the two 1C units of purines (ATP, ADP, AMP, GTP, GDP, GMP) from [$^{13}$C]-formate, the purine mass isotopomer fractions were first corrected by the natural isotope abundance. The remaining M + 1 and M + 2 fractions are due to

incorporation of $^{13}$C from formate at the two 1C units in purines. Assuming that both carbons can be labeled with equal probability $p$, the purines M + 1 and M + 2 fractions are modeled by the equations $P_1 = (1-P_0)2p(1-p)$ and $P_2 = (1-P_0)p^2$, respectively, where $P_0$, $P_1$, and $P_2$ are the purines M + 0,1,2 fractions, respectively. Solving these equations for $p$ we obtain $P1C = p = 2P_2/(P_1 + 2P_2)$. The reported $^{13}$C enrichment in the one-carbon units of purines was obtained using the latter equation and the measured $P_1$ and $P_2$.

**In vivo relative rate of serine catabolism to formate (SCF)**. To estimate the relative rate of in vivo serine catabolism to formate we postulated a metabolic flux analysis (MFA) model whereby the formate incorporated into purines at a given tissue comes from the serine one-carbon catabolism at that tissue or from plasma formate. Within this model the tissue M + 1 P1C ($P1C_1$) is given by the equation $P1C_1 = sS_1 + (1-s)F_1$, where $s$ is the relative rate of serine one-carbon catabolism (i.e., SCF), $S_1$ is the fraction of tissue M + 1 serine and $F_1$ is the fraction of plasma M + 1 formate. Solving the latter equation for $s$ we obtain $s = (P1C_1 - F_1)/(S_1 - F_1)$. In the liver we have the additional contribution of methanol conversion to formate, resulting in $P1C_1 = sS_1 + (1-s-m)F_1 + m$, where $m$ is the relative rate of methanol conversion to formate. Because this system has now two independent variables ($s$ and $m$) and only one equation, it is undetermined and we cannot determine $s$ in the liver. Our method works even when the in tissue conversion of $^{13}$C–MeOH to formate is not taken into account. The reason why it works is that the method is based on the difference between the $^{13}$C-enrichment in tissue serine, plasma formate and one-carbon units in purines (P1C). Since $^{13}$C–MeOH is the source of $^{13}$C-formate, in situ conversion of $^{13}$C–MeOH to $^{13}$C-formate will tend to make the $^{13}$C-P1C closer to the plasma $^{13}$C-formate. Therefore, the observation that $^{13}$C-P1C is different from plasma $^{13}$C-formate, is evidence that there is catabolism of serine to formate. The larger is the difference, the larger the relative contribution of the serine catabolism to formate. In a strict sense, the reported relative rate of serine catabolism to formate is a lower bound to the actual value. That said, the serine catabolism to formate could be even higher than what we report.

**In vivo relative fraction of de novo synthesized purines**. To estimate the fraction of de novo synthesized purines ($p$) we took into account that the fraction of M + 0 purines is given by purines remaining from the pre-existing pool at the start of the experiment $(1-p)$ plus de novo synthesized purines with unlabeled one-carbon units: $P_0 = 1 - p + p[1-(1-P1C_1)^2]$. Solving this equation for $p$ we obtain our working equation to estimate $p$ given experimental measurements of $P_0$ and our previous estimate of $P1C_1$: $p = (1-P_0)/(1-P1C_1)^2$.

**Statistics**. Data was first assessed to be eligible for respective statistical test. For pairwise comparison an unpaired $t$-test with Welch's correction was applied using the GraphPad Software. The statistical significance of PCC was estimated using a permutation test with 1,000,000 permutations. For normalization of invasion data, within one experiment, the mean value of the ten technical replicates (two chambers with five images per chambers) per condition was calculated and divided by the global mean of the experiment.

**Data availability**. The authors declare that all the data supporting the findings of this study are available within the article and its Supplementary Information Files and/or from the corresponding author upon reasonable request.

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

## Acknowledgements

This work was supported by Cancer Research UK C596/A21140. J.M. was supported by a DFG Fellowship Grant Number: ME 4636/2-1. We acknowledge the Cancer Research UK Glasgow Center (C596/A18076) and the BSU facilities at the Cancer Research UK Beatson Institute (C596/A17196). A.S. was supported by Fondation Cancer Luxembourg (INVGBM project). NCH601 cells were generated in the laboratory of Dr Christel Herold-Mende (Department of Neurosurgery, University of Heidelberg). J.M., A.S. and S.N. thank Vanessa Barthelemy for technical assistance with the shRNA cell lines.

## Author contributions

J.M., K.O., A.S. performed in vitro experiments. J.V.V., D.A., G.B.B., N.W., S.D., J.P.M. and E.D. performed in vivo experiments. J.M. and M.P. processed tissues from in vivo experiments and analyzed data. K.J.P. and K.B. participated in the design of and supervised in vivo experiments. S.N. supervised the in vitro invasion experiments and shRNA knockdown generation. A.V. designed computer scripts and analyzed data. J.M. and A.V. designed the project and wrote the manuscript. All authors read the manuscript.

## Additional information

**Competing interests:** The authors declare no competing interests.

