## [Peer Review File · Nature Communications]

Reviewers' comments:

Reviewer #1 (Remarks to the Author):

The authors have addressed my concerns. New data are provided on invasiveness with formate addition. The authors may want to back off the claims about formate overflow functioning to promote invasion, as these data are preliminary (and mitochondrial production is different than endogenous treatment). Nevertheless, they provide some potential functionality which can be investigated further in the future.

Reviewer #2 (Remarks to the Author):

In the manuscript entitled "Increased formate overflow is a hallmark of cancer" Meiser and colleagues addressed some of the major concerns raised in the previous revision of the manuscript.

The work is now more focused on determining the correlation between the redox state and formate overflow in cell lines and in vivo. However, the authors need to perform some additional controls and clarify some points in order to make this work more convincing.

1) In their response to the concern about novelty, the authors claim that "all previous work on formate release by cancer cells is limited to in vitro data." However, in their recent Science Advances work, the authors actually determined a correlation between serine and formate levels in WT and in APCmin mice upon labelling with ^{13}C -serine. Therefore, the authors' statement is not entirely accurate. Furthermore, this referee feels that the previous experiments were actually more appropriate to determine formate overflow than the methanol tracing presented in the current manuscript. Indeed, as defined by the authors, formate overflow entails the extracellular release of formate, rather than serine catabolism (as measured by methanol tracing). The authors need to better clarify the scope of these labelling experiments in the context of formate overflow, or to define in a more broader way the concept of "overflow" (which is actually rather surprisingly used even for lactate in the introduction)

2) The authors also address the concern related to the measurement of NADH/NAD by providing a titration of these two species by MS. However, the concern was not about the ability of the MS platform to measure them but, rather, the fact that the processing of the samples and could affect the oxidation state of these molecules, generating a ratio that is not a genuine reflection of the actual redox state of the cell. The difficulty of extracting these molecules is noticed even by the authors in the PyMT model, where they found that NADH levels were not detectable, allowing them to conclude that the tissue is highly oxidative. Of note, due to this technical issue, they had to modify the calculation of the SCI. However, whether this is biological or a technical issue is unclear. Since the determination of NAD/NADH ratio is crucial for this work, it is advisable to back up the MS measurement with orthogonal assays of choice. These experiments would show that the extraction method for MS preserves these two species in their original state (i.e. that further redox reactions do not occur during extraction or once all metabolites are in solution). Finally, the authors need to show NAD⁺ and NADH values (rather than ratio) and a titration experiment, to demonstrate the dynamic range of the measurements made by LC-MS and, more importantly, to show that the quantifications done for the various experiments are within that dynamic range.

3) The authors manipulated the redox state of the cells by pharmacological agents and showed

that formate release is increased when NAD/NADH is increased. In the attempt to generalise these findings, the authors present experiments where cells were genetically modified to ablate SHMT1/2. The observation that the ablation of SHMT2 blunts formate release whilst increasing NAD/NADH ratio suggest that formate overflow depends by and large on mitochondrial 1-carbon metabolism. This interpretation seems to be supported by the MTX experiments, where the increase in formate overflow is coupled by a (modest) increase in respiration. Unless other ways to manipulate cytosolic redox state are presented, the authors' hypothesis that it is the general redox state to support formate overflow (rather than mitochondrial function alone) is not supported. Furthermore, the authors need to present OCR values for the SHMT1/2-deficient cells to complete the study.

4) The new functional experiments on the role of formate in cell migration are interesting but this referee believes that they are misconceived and do not provide an explanation as to why formate overflow is important for cancer cell (as claimed by the authors). Indeed, these experiments imply a non-cell autonomous role of formate, rather than a selective advantage for the cancer cell itself. These experiments also imply that formate is taken up by cancer cells (rather than released) to promote invasion.

The rescue experiment in MTHFDL1 is even more puzzling. In normal conditions, cancer cells produce and release formate via the mitochondrial pathway. If this enzyme is silenced, less formate is released and invasion is decreased. This is potentially an interesting finding. However, the observation that exogenous formate rescues the migration defects indicate that the intracellular use of formate, rather than its release is important for this invasion. I do not think that the authors can conclude that formate overflow promotes glioma cell invasion as a cell-autonomous process. This part of the work would need to be better framed. For instance, instead of claiming that formate overflow drives cell migration, the authors could make a case for a non-cell autonomous effect of this molecule in the tumour microenvironment.

5) Finally, the referee appreciates the extended explanation of the ¹³C-methanol experiment. However, the presentation of measurement of intracellular ¹³C-formate in the tissues investigate would corroborate their method.

6) In Fig1.b the R-squared and P value should be included.

7) The description of some of the experiments, including the methanol labelling is cumbersome. I would suggest simplifying some of the most technical parts of the work to make it more accessible to readers.

RESPONSE TO REVIEWERS' COMMENTS:

Reviewers' comments:

Reviewer #1 (Remarks to the Author):

The authors have addressed my concerns. New data are provided on invasiveness with formate addition. The authors may want to back off the claims about formate overflow functioning to promote invasion, as these data are preliminary (and mitochondrial production is different than endogenous treatment). Nevertheless, they provide some potential functionality which can be investigated further in the future.

Response: We are delighted that the Reviewer's concerns have been addressed in our revised manuscript. We also agree that the invasion hypothesis needs further research in the future and we modified the interpretation of the data accordingly.

Reviewer #2 (Remarks to the Author):

In the manuscript entitled "Increased formate overflow is a hallmark of cancer" Meiser and colleagues addressed some of the major concerns raised in the previous revision of the manuscript.

Response: We are happy to hear that major concerns of the Reviewer have been addressed in our revised manuscript.

The work is now more focused on determining the correlation between the redox state and formate overflow in cell lines and in vivo. However, the authors need to perform some additional controls and clarify some points in order to make this work more convincing.

1) In their response to the concern about novelty, the authors claim that "all previous work on formate release by cancer cells is limited to in vitro data." However, in their recent Science Advances work, the authors actually determined a correlation between serine and formate levels in WT and in APCmin mice upon labelling with ¹³C-serine. Therefore, the authors' statement is not entirely accurate. Furthermore, this referee feels that the previous experiments were actually more appropriate to determine formate overflow than the methanol tracing presented in the current manuscript. Indeed, as defined by the authors, formate overflow entails the extracellular release of formate, rather than serine catabolism (as measured by methanol tracing). The authors need to better clarify the scope of these labelling experiments in the context of formate overflow, or to define in a more broader way the concept of "overflow" (which is actually rather surprisingly used even for lactate in the introduction)

Response: To reply to this point I split my response in two parts:

i) Regarding the novelty:

In our previous study¹, our investigations were limited to whole body catabolism of serine to formate in healthy mice. As we have stated in our previous rebuttal letter, these experiments did not provide any experimental evidence that tumors have increased rates of serine catabolism with

formate overflow *in vivo*. In this submission, we now provide experimental evidence that some tumors have the highest rate of serine catabolism of all tissues analysed. As far as we know this knowledge is novel and has not been published elsewhere. Knowing that increased formate overflow is a metabolic feature of some tumors can represent a target for therapeutic intervention in the future. The next major question is to understand the function of this phenomenon. With our invasion data, we provide first evidence into that direction which we will follow up in the future (for further details please read our answer to concern 4 below). In that respect, the reported correlation with formate and cancer cell invasion can make this publication to a landmark paper as there is (to our knowledge) no reported evidence in that direction until now.

ii) Regarding the MeOH tracing:

While we acknowledge that the Reviewer appreciates our previous *in vivo* experiments to analyse plasma formate levels, we disagree that these approaches would be suitable “to determine formate overflow *in vivo*”. “To determine”, we need to estimate the actual rate *in vivo*. To analyse the rate of serine catabolism *in vivo* and to compare different tissues to each other, we need to apply stable isotope tracing. This approach is an absolute requirement to answer the question if tumors have increased rates of serine catabolism *in vivo* compared to the adjacent tissue and other tissues of the organism. A prerequisite for metabolic flux analysis (required to determine the actual **rate** of a metabolic pathway) is isotopic steady state². The beauty of MeOH tracing is that we reach isotopic steady state with a single intraperitoneal injection. We agree that each method has its advantages and disadvantages, but we feel that, to answer the here addressed question, our approach is the most suitable one. By determining P1C, we determine the labelling pattern of the tissue formate pool and thus we obtain an estimate of formate production from serine and thus the rate of serine catabolism **to formate** (for technical details we refer to our method section).

Finally, we would like to define our understanding of overflow metabolism. We define this metabolic state as a metabolic stress condition where organic molecules follow incomplete catabolism with the subsequent excretion of products thereof that could otherwise be fully oxidised or used for biomass production. Overflow metabolism applies to the phenomenon of lactate excretion as well as to formate excretion.

2) The authors also address the concern related to the measurement of NADH/NAD by providing a titration of these two species by MS. However, the concern was not about the ability of the MS platform to measure them but, rather, the fact that the processing of the samples and could affect the oxidation state of these molecules, generating a ratio that is not a genuine reflection of the actual redox state of the cell. The difficulty of extracting these molecules is noticed even by the authors in the PyMT model, where they found that NADH levels were not detectable, allowing them to conclude that the tissue is highly oxidative. Of note, due to this technical issue, they had to modify the calculation of the SCI. However, whether this is biological or a technical issue is unclear. Since the determination of NAD/NADH ratio is crucial for this work, it is advisable to back up the MS measurement with orthogonal assays of choice. These experiments would show that the extraction method for

MS preserves these two species in their original state (i.e. that further redox reactions do not occur during extraction or once all metabolites are in solution). Finally, the authors need to show NAD⁺ and NADH values (rather than ratio) and a titration experiment, to demonstrate the dynamic range of the measurements made by LC-MS and, more importantly, to show that the quantifications done for the various experiments are within that dynamic range.

Response: We do not understand the reviewer request to provide extensive validation for our LC-MS protocol to quantify NAD⁺ and NADH. This protocol has been extensively used by us and others in preceding publications. Furthermore, it has been validated by our Mass Spectrometry platform team and was published in the here cited paper³. After discussing the Reviewers concerns with our platform team we do not see any additional need to validate our approach.

1. To back-up our viewpoint we want to refer to three (out of many) publications where similar approaches have been performed. The first paper is published by Josh Rabinowitz a leading expert in mammalian metabolism and LC-MS analysis in *Nature Chemical Biology*. In the here cited paper⁴ the NAD/NADH ratio was also used as a robust readout for the cell's redox state. Similar to our metabolite extraction approach the authors used an acetonitrile/methanol/water mix.

The second and even more important publication is from our own institute using the very same Mass Spectrometry platform that we used for our study. We applied the same extraction protocols and used the same machines with the same analytical protocol. In the work by Cardaci et al⁵, published in *Nature Cell Biology*, the NAD/NADH ratio was also used as a readout (Figure 2d). This peer-reviewed high impact publication demonstrates that our protocols are established and suitable for our work.

Towards the suggested "orthogonal" approach we want to refer to the work by Quinn et al⁶. The authors used an imaging approach and a LC-MS based approach to determine NAD⁺ and NADH levels as well the respective ratio. They demonstrated that both approaches correlated in a linear relation which also implies that the applied methods to determine the NAD⁺/NADH ratio using LC-MS was appropriate. As a matter of fact, the LC-MS quantification is often used as the gold standard to validate methods based on other technologies.

Additionally, our data clearly shows that SCF and SCI complement each other. It is important to note that, to calculate SCF, no NAD⁺/NADH ratio is needed. SCF estimation is solely based on ¹³C enrichment in different metabolites.

Overall, we do not see the point here to invest additional resources to "reinvent the wheel" for an established methodology.

2. In the PyMT mice we observed that within the mammary gland the NADH levels are extremely low which is why we could not infer the before used ratio. The reason for this is not given by a "technical difficulty" but by the tissue specific metabolic profile. We note that for those same tissues NAD⁺ was detected in significant amounts, indicating that NAD is mostly present in its oxidized form. Moreover, in this very same experiment we detected NADH in the liver, brain and spleen (as in the other experiments before) (supplementary figure 6) and only in the mammary gland we could not detect it, indicating that this is a biological feature of the tissue rather than a technical issue. If in this particular experiment it would have been caused by technical issues then also the NADH detection in the other tissues should have been affected. In any case, to circumvent this technical hurdle (caused by a biological feature) of having very low NADH, we changed our redox readout from NAD⁺/NADH to NAD⁺/(NADH+NAD⁺), as a mean to obtain meaningful, reliable and robust results when NADH is low.

3. Finally, we want to stress that all of our results are based on several independent experiments. Also for the *in vivo* experiments, our tissue specific characterisation of the NAD⁺/NADH ratio across different experiments (as given by the relative association of the NAD⁺/NADH ratio between brain, liver and spleen) is robust and reproducible. **IF** we could not properly quench the metabolism quick enough, we would obtain much higher variation in relative standard errors within each group. This would be reflected in the data, and differences between conditions and/or tissues could not be revealed. In the latter case, variation would be given by random phenomenon rather than reflected by underlying biological features. In this case, our results would not be reproducible between the performed independent experiments.

3) The authors manipulated the redox state of the cells by pharmacological agents and showed that formate release is increased when NAD/NADH is increased. In the attempt to generalise these findings, the authors present experiments where cells were genetically modified to ablate SHMT1/2. The observation that the ablation of SHMT2 blunts formate release whilst increasing NAD/NADH ratio suggest that formate overflow depends by and large on mitochondrial 1-carbon metabolism. This interpretation seems to be supported by the MTX experiments, where the increase in formate overflow is coupled by a (modest) increase in respiration. Unless other ways to manipulate cytosolic redox state are presented, the authors' hypothesis that it is the general redox state to support formate overflow (rather than mitochondrial function alone) is not supported. Furthermore, the authors need to present OCR values for the SHMT1/2-deficient cells to complete the study.

Response: As requested by the Reviewer, we include now new data on oxygen consumption rates by wild-type and SHMT2 deficient cells (as supplementary figure 3). Compared to parental WT cells, we did not observe a significant change in OCR in the analysed cell lines (Figure 1 below). Interestingly, since in the SHMT2 KO cells NAD is more oxidized (higher NAD^+/NADH , Figure 1p of our manuscript) while the mitochondrial activity does not change significantly, this represents a scenario where the cell redox state is affected without changes in mitochondrial respiration. Here, as in the MTX treated condition, we think that growth inhibition reduces the demand for electron acceptors. That together with no changes in the cells oxidizing capacity results in a more oxidized state. In summary, the redox state, as any other metabolic variable, can be altered by changes in both demand and supply. At constant oxidation by mitochondrial activity, a reduction in the electron acceptor demand is another mean to change the cell redox state.

Figure 1: Oxygen consumption rate of HAP1 cells: WT and SHMT2 KO. Each dot indicates one independent experiment (n=5). Error bars indicate s.e.m.. There is no statistical significant difference between conditions.

4) The new functional experiments on the role of formate in cell migration are interesting but this

referee believes that they are misconceived and do not provide an explanation as to why formate overflow is important for cancer cell (as claimed by the authors). Indeed, these experiments imply a non-cell autonomous role of formate, rather than a selective advantage for the cancer cell itself. These experiments also imply that formate is taken up by cancer cells (rather than released) to promote invasion.

The rescue experiment in MTHFDL1 is even more puzzling. In normal conditions, cancer cells produce and release formate via the mitochondrial pathway. If this enzyme is silenced, less formate is released and invasion is decreased. This is potentially an interesting finding. However, the observation that exogenous formate rescues the migration defects indicate that the intracellular use of formate, rather than its release is important for this invasion. I do not think that the authors can conclude that formate overflow promotes glioma cell invasion as a cell-autonomous process. This part of the work would need to be better framed. For instance, instead of claiming that formate overflow drives cell migration, the authors could make a case for a non-cell autonomous effect of this molecule in the tumour microenvironment.

Response: We agree with the Reviewer that whether formate overflow promotes invasion requires more discussion. What we can say with confidence is that high rates of serine catabolism to formate production, in excess of the biosynthetic demand, promote invasion. We need to specify in excess of the biosynthetic demand because knockdown of MTHFD1L does not inhibit proliferation, while it inhibits invasion. We also note that once formate is produced in excess of the biosynthetic demand it needs to find an outlet, and this outlet is formate overflow. We adapted the text in the discussion section to reflect this more precise interpretation of our data.

5) Finally, the referee appreciates the extended explanation of the ¹³C-methanol experiment. However, the presentation of measurement of intracellular ¹³C-formate in the tissues investigate would corroborate their method.

Response: We appreciate that our extended explanations increased the understanding by the reader. In fact, we tried different approaches to measure formate in tissue. However, this is technically very challenging because of the nature of the molecule. To measure this very small and volatile compound it needs to be derivatised and can then be analysed with GC-MS. However, the derivatisation has to occur in solution because freeze-drying the sample would result in sample loss (due to the volatile nature of formate). To maintain the metabolic state, we need to constantly work in cold (freezing) conditions. To keep our extraction solvent liquid, it contains methanol. The problem with the alcohol and its hydroxyl group is that it is also derivatised by the MCF and that it traps all the derivatisation agent. We searched the literature and tested different approaches but until now we did not establish a relying protocol to accurately quantify formate in tissues. Therefore, we make use of P1C which is a direct readout of the formate labelling pool. This methodology has already successfully been applied in our recent publication in *Nature*⁷.

6) In Fig1.b the R-squared and P value should be included.

Response: These values were already given in the text, we included them now also in the figure.

7) The description of some of the experiments, including the methanol labelling is cumbersome. I would suggest simplifying some of the most technical parts of the work to make it more accessible to readers.

Response: We thank the Reviewer for this suggestion and we adapted the main text accordingly. However, we feel that a part of the novelty in this publication is the new methodology that we developed to obtain the results. In that respect we want to keep some technical details in the main

text as there might also be a group of readers that is specifically interested in that particular method. But we now try to keep it to a minimum.

- 1 Meiser, J. *et al.* Serine one-carbon catabolism with formate overflow. *Sci Adv* **2**, e1601273, doi:10.1126/sciadv.1601273 (2016).
- 2 Buescher, J. M. *et al.* A roadmap for interpreting (13)C metabolite labeling patterns from cells. *Curr Opin Biotechnol* **34**, 189-201, doi:10.1016/j.copbio.2015.02.003 (2015).
- 3 Mackay, G. M., Zheng, L., van den Broek, N. J. & Gottlieb, E. Analysis of Cell Metabolism Using LC-MS and Isotope Tracers. *Methods Enzymol* **561**, 171-196, doi:10.1016/bs.mie.2015.05.016 (2015).
- 4 Bennett, B. D. *et al.* Absolute metabolite concentrations and implied enzyme active site occupancy in Escherichia coli. *Nat Chem Biol* **5**, 593-599, doi:10.1038/nchembio.186 (2009).
- 5 Cardaci, S. *et al.* Pyruvate carboxylation enables growth of SDH-deficient cells by supporting aspartate biosynthesis. *Nat Cell Biol* **17**, 1317-1326, doi:10.1038/ncb3233 (2015).
- 6 Quinn, K. P. *et al.* Quantitative metabolic imaging using endogenous fluorescence to detect stem cell differentiation. *Sci Rep* **3**, 3432, doi:10.1038/srep03432 (2013).
- 7 Burgos-Barragan, G. *et al.* Mammals divert endogenous genotoxic formaldehyde into one-carbon metabolism. *Nature* **548**, 549-554, doi:10.1038/nature23481 (2017).

REVIEWERS' COMMENTS:

REVIEWER#2

Editorial Note:

Reviewer#2 provided only confidential comments to the editors.

Reviewer#2 expresses his disappointment for the refusal to confirm the NAD/NADH measurements using an enzymatic assays but as you appear confident in your method (on the basis of the cited publications) and take full responsibility for it support proceeding with the publication. Reviewer#2 also highlights that the role of formate release as inducer of cell migration must be better framed as it still does not address the question of why and if the release of formate is important for the cancer cell itself rather than being a non-cell autonomous consequence of formate overflow.

RESPONSE TO REVIEWERS' COMMENTS:

REVIEWER#2

Reviewer#2 expresses his disappointment for the refusal to confirm the NAD/NADH measurements using an enzymatic assay but as you appear confident in your method (on the basis of the cited publications) and take full responsibility for it support proceeding with the publication.

Response: We would like reiterate that the quantification of NAD⁺ and NADH by LC-MS is, to our knowledge, the best and most reliable method currently available to quantify these metabolites in cell and tissue extracts. Furthermore, our metabolomics core laboratory has extensive experience quantifying NAD⁺ and NADH using LC-MS. In the case of tissue extracts, where the time lapse associated with tissue retrieval may be an issue, we provide NAD⁺ and NADH quantifications for the adjacent non-cancer tissue as a control for the quantifications in cancer tissues. Both the cancer and non-cancer tissue are subject to the same processing steps. Therefore we are confident about our report that the murine cancer tissues profiled have higher NAD⁺/NADH ratios than their corresponding adjacent non-cancer tissue.

Reviewer#2 also highlights that the role of formate release as inducer of cell migration must be better framed as it still does not address the question of why and if the release of formate is important for the cancer cell itself rather than being a non-cell autonomous consequence of formate overflow.

Response: We agree with the reviewer that our data is not sufficient to distinguish between intracellular and extracellular formate as the signal promoting invasion. We have modified the discussion around this point.